# Chlorogenic acid inhibits NLRP3 inflammasome activation through Nrf2 activation in diabetic nephropathy

**Liping Bao**[1]*, **Yuhan Gong**[1], **Wenji Xu**[2], **Jun Dao**[1], **Jinjin Rao**[1], **Haihui Yang**[1]*

**1** Department of Nephrology, Pu'er People's Hospital, Pu'er, Yunnan, China, **2** Department of Gastrointestinal and Plastic Surgery, Pu'er People's Hospital, Pu'er, Yunnan, China

* bao1506395676@163.com (LB); 1506395676@qq.com (HY)

## Abstract

Diabetic nephropathy (DN) is the single largest cause of end-stage renal disease (ESRD). Inflammation reaction mediated by NLRP3 inflammasome and Nrf2-related oxidative stress have been considered to play a very important role in the progress of diabetic nephropathy (DN). Effective drugs for the treatment of diabetic nephropathy still need to be explored. Chlorogenic acid (CGA) is a kind of polyphenol with a Nrf2 activation property widely existed in nature. The aims of this study were to evaluate the renoprotective effect of CGA and to elucidate the anti-inflammation mechanisms involved. In the present study, we established a diabetic rat model to investigate the renoprotective effect of CGA in vivo. The results show that the level of serum creatinine (Scr), blood urea nitrogen (BUN), and urinary protein excretion in diabetic rats were significantly decreased after CGA intervention. CGA administration can active the Nrf2 pathway and inhibit NLRP3 inflammasome activation. Notably, Nrf2 siRNA transfection nullified the inhibitory effects of CGA on NLRP3 inflammasome activation in vitro. To summarize, our present study provided evidence that chlorogenic acid can slow the progression of diabetic nephropathy progression, and the effect is associated with suppression of NLRP3 inflammasome activation via through modulation of the Nrf2 pathway, suggesting its therapeutic implications for diabetic nephropathy.

## 1. Introduction

Diabetic nephropathy (DN), one of the most threatening complications of patients with diabetes, has become the leading cause of end-stage renal disease (ESRD) worldwide [1]. DN is characterized by proteinuria and the progressive decline in renal function. Numerous studies have shown that inflammation reaction plays a certain part in the development of DN [2,3].

Reports found that inflammation reaction mediated by NOD-like receptor protein 3 (NLRP3) inflammasome was responsible for the DN progression. NLRP3 inflammasome is composed of NLRP3, caspase-1 and apoptosis-associated speck-like protein containing a C-terminal caspase recruitment domain (ASC) [4]. In the diabetic state, NLRP3 inflammasome is activated, then NLRP3 recruits ASC and cleaves caspase-1. Next, cleaved caspase-1 promotes

**Funding:** The present study was supported by Yunnan Provincial Science and Technology

Department (202101AY070001-304) and Yunnan Provincial Health Commission(D-2018021 to HY).

the conversion of inflammatory factors such as pro-IL-1β and pro-IL-18 into IL-1β and IL-18 and released outside the cell, which leading to pro-inflammatory responses [5]. Excessive activation of NLRP3 inflammasome may lead to uncontrolled inflammation and cell death. Hence, therapies targeting activation of NLRP3 inflammasome may play an effective role in protecting renal function and delay the progression of DN. Oxidative stress is one of the fundamental causes of chronic complications of diabetes. Nuclear factor erythroid-derived 2-related factor 2 (Nrf2) takes on a critical part in the defense system against oxidative stress. In response to cellular oxidative stress injury, Nrf2 evades repression by Kelch-like epichlorohydrin-associated protein 1 (Keap1) and accumulates within the nucleus, and then activates the transcription of phase II detoxifying anti-oxidant enzymes, including heme oxygenase-1 (HO-1) [6]. Recent studies have shown that activating the Nrf2 pathway could inhibit the activation of NLRP3 inflammasome and significantly reduced the expression of downstream factors caspase-1, IL-18, and IL-1β in acute liver injury, cerebral artery occlusion-reperfusion injury and severe lupus nephritis [7–9]. We hypothesized that Nrf2 may negatively regulated the activation of NLRP3 inflammasome in DN.

Chlorogenic acid (CGA), a phenolic compound, is widely found in Chinese herbal medicines, such as *Flos lonicerae* and *Eucommia ulmoides* [10,11]. Recent studies have showed that CGA possesses various pharmacological activities, including anti-inflammatory [12], antioxidant [13], antibacterial activities [14], and anticarcinogenic [15]. Moreover, several recent experiments indicated that CGA treatment attenuated acute liver and lung impairments by hindering NLRP3 inflammasome activation [16,17]. However, the effect of CGA on NLRP3 inflammasome and the underlying mechanisms in DN are still unclear. Our previous study has shown that pre-treatment with CGA increased the renal expression of Nrf2 and the downstream target heme oxygenase-1 (HO-1) [18]. The aim of the present study was to explore the effect of CGA on NLRP3 inflammasome in the high-fat diet (HFD)/streptozotocin (STZ)-induced diabetic rat kidney and human proximal tubule epithelial cell line HK-2 and to further explore the potential connection between the Nrf2 pathway and NLRP3 inflammasome.

## 2. Materials and methods

### 2.1. Reagents

Chlorogenic acid (CGA) was purchased from MACKLIN Company (Shanghai, China). Biochemical kits for serum creatinine (Scr), blood urea nitrogen (BUN) and urinary protein were purchased from Changchun Huili Company (Changchun, China). The anti-NLRP3(DF7438), anti- cleaved caspase-1(AF5418), anti- IL-1β(AF5103), anti- IL-18(DF6252), and anti- HO-1 (AF5393) antibodies were purchased from Affinity (Melbourne, Australia). Anti- Nrf2(16396-1-AP) and anti-H3(17168-1-AP) antibodies were purchased from Wuhan Sanying Biotechnology Co., Ltd (Wuhan, China).

Fetal bovine serum (FBS) and dulbecco's modified Eagle's medium/F12 (DMEM/F12) medium were obtained from HyClone (Logan, UT, USA).

### 2.2. Animals and drug treatments

A total of 18 male specific pathogen-free (SPF) grade Wistar rats (aged 6–8 weeks, 180–200 g of body weight) were obtained from the Disease Prevention and Control Centers of Hubei Province. All animal experiments were conducted according to a protocol approved by Animal Care and Use Committee of the Center for Disease Prevention and Control in Hubei Province (No. 202220148). After 1 week of adaptive feeding, the rats were randomly selected and divided into three groups (6 rats in each group): a normal control group (NC), type 2 diabetes mellitus model group (DM), and DM+CGA group. The DM and DM + CGA groups were fed

a high-fat diet (D12492, Hua Fu Kang Biotechnology Co., Ltd. Beijing, China) for 4 weeks to cause insulin resistance. And then all of the insulin resistance rats were injected with low-dose STZ (35 mg/kg, St. Louis, MO, USA) dissolved in citrate buffer solution once. The DM and DM + CGA groups were continued to be fed a high-fat diet until the end of the experiment. The successful establishment of the diabetes model was confirmed on the 3rd day after STZ injection by measuring the blood glucose level greater than 16.7 mmol/L. The rats in DM + CGA group were intraperitoneally injected with CGA for 10 weeks at a dose of 10 mg/kg/d. Rats in the NC and DM groups were intraperitoneally injected with equivoluminal distilled water.

## 2.3. Blood and urine chemistries analysis

After 10 weeks, animals were sacrificed with intraperitoneal injection of 1% pentobarbital solution, and if animals show signs of pain, increase the dosage appropriately. Serum creatinine, BUN and urinary protein excretion were measured using biochemical kits according to the manufacturer's instructions.

## 2.4. Histopathology

The kidney tissues were fixed in 4% paraformaldehyde and embedded in paraffin. The embedded kidney tissues were cut into 3-μm sections. The tissue sections were subsequently stained with hematoxylin and eosin (HE) and periodic acid- Schiff (PAS).

## 2.5. Immunohistochemistry

After deparaffinization and hydration, the slices were subjected to antigen heat repair using an electric pottery stove for 15 minutes. To block endogenous peroxidase, the sections were incubated with 3% $H_2O_2$ for 15 minutes. Then the sections were blocked with goat serum for 30 minutes. Subsequently, the sections were incubated with primary antibodies (1:100) overnight at 4˚C. Washed with phosphate-buffered saline (PBS) for three times, the sections were incubated with horseradish peroxidase (HRP)-labelled secondary antibodies for 30 minutes, then stained with 3,3'-diaminobenzidine (DAB) and re-stained with haematoxylin. The sections were observed by a microscope and analyzed Image J software. The protein expression levels were reflected with the average optical density (AOD) values.

## 2.6. Cell culture and stimulation

Human proximal tubule epithelial (HK-2) cells were obtained from Nanjing KeyGen Company(Nanjing, China).The cells were cultured in low concentration of glucose (5.6 mmol/l) DMEM/F12 supplemented with 10% fetal bovine serum (FBS), 100 μg/mL streptomycin and 100 U/mL penicillin at 37˚C in an incubator with 5% CO2. Cultured cells were divided into five groups: the normal control group (NC, 5.6 mmol/L glucose); the high concentration of glucose group(HG,30 mmol/L glucose); the different concentrations of CGA(20, 50 and 100 μM) +30 mmol/L glucose group(CGA + HG). After 48 h of CGA and HG treatment, HK-2 cells were collected for subsequent experiments.

## 2.7. siRNA transfection

Predesigned small interfering RNA (siRNA) targeting Nrf2 and negative controls siRNA were obtained from Guangzhou Ribo Company. (Guangzhou, China). HK-2 cells were transfected with control or Nrf2 sense siRNA using Lipofectamine[TM] 2000 transfection reagent. HK-2 cells were seeded onto 6-well culture plates in serum-free OPTI-MEM medium (Invitrogen,

Carlsbad, CA, USA). When the cells at 60–80% confluence, they were incubated with 50 nM control or Nrf2 siRNA. After 6 h incubation, the cells were incubated with high concentration glucose medium alone or administrated with CGA (100 μM) for an additional 48 hours before being harvested for the further experiments.

## 2.8. Western blot analysis

Total proteins were extracted from kidney tissues and cultured cells, and RIPA buffer was used for the sample suspension. Protein concentration of cells was determined with bicinchoninic acid (BCA) protein assay kit (ASPEN, USA). 30 μg proteins in mixture solution were separated by 10% sodium dodecyl sulfate-polyacrylamide gel electrophoresis gel and then electrophoretically transferred to PVDF membranes of 0.45 μm. After being blocked with 5% skim milk in Tris-buffered saline (TBS) containing 0.1% Tween-20 (TBST) for one hour, the blocked membranes were incubated separately with diferent kinds of primary antibodies at 4°C overnight. The PVDF membranes were washed 3 times by TBST for 10 minutes each time and subsequently incubated with horseradish peroxidase-conjugated secondary antibody for two hours at room temperature. The protein bands were detected by the chemiluminescence method. The relative protein level of each group was analyzed by the IPP analysis software.

## 2.9. Statistical analysis

All experimental data are based on three biological replicates. The data were analyzed using GraphPad Prism 12 software. All data are presented as the means ± SD. Differences between two groups were conducted by t-test. It was considered statistically significant when the value of $P < 0.05$.

# 3. Results

## 3.1. CGA alleviated albuminuria and improved renal function in high-fat diet (HFD)/ streptozotocin (STZ) -induced diabetic rats

As shown in Fig 1 kidney weight of diabetic rats was significantly higher than normal rats, diabetic rats treated with CGA had lower kidney weight than untreated diabetic rats. However, body weight of diabetic rats was significantly lower than normal rats, diabetic rats treated with CGA had higher body weight than untreated diabetic rats. In addition, the DM group had significantly increased Scr and BUN concentrations and urinary protein excretion compared with the control group. CGA treatment reduced Scr and BUN concentration and urinary protein excretion. These data showed that CGA protects the kidneys in diabetic rats. Specific values are in S1 Table.

   A. Kidney weight. B.Body weight. C. Urinary protein. D. serum creatinine (Scr). E. Blood urea nitrogen (BUN). *$P < 0.05$ vs. NC group, **$P < 0.01$ vs. NC group, #$P < 0.05$ vs. NC group, ##$P < 0.01$ vs. DM group.

## 3.2.CGA attenuated renal histopathological injury in high-fat diet (HFD)/ streptozotocin (STZ) -induced diabetic rats

The changes in renal histopathology are shown in Fig 2. HE and PAS staining staining showed that diabetic rats had notable glomerular hypertrophy, mesangial matrix deposition (red arrow), renal tubular vacuolar degeneration (black arrow) and renal tubular dilatation (green arrow). 10 weeks of treatment with CGA significantly improved these pathological changes. The results indicated that CGA attenuated renal morphological abnormalities in diabetic rats.

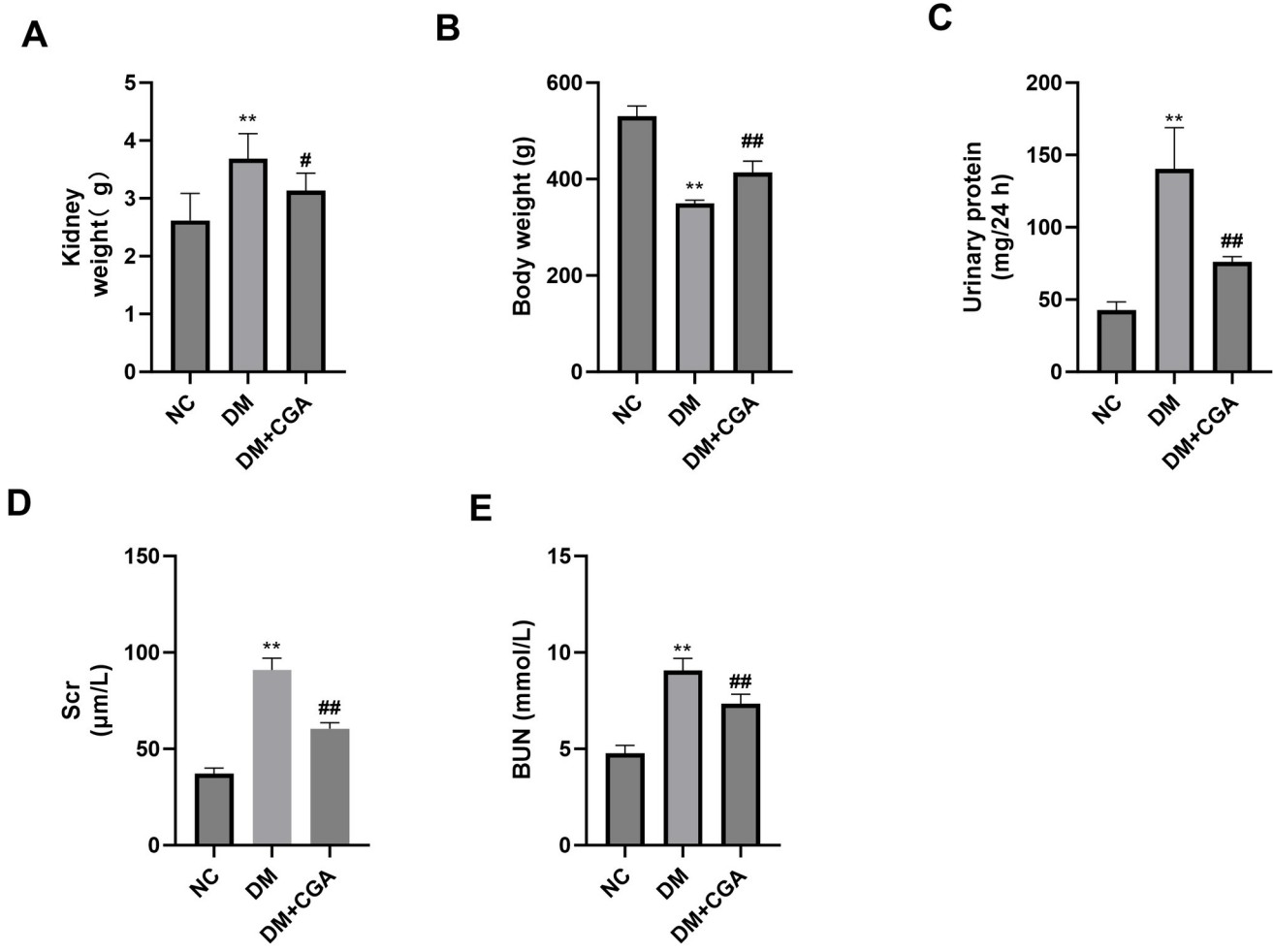

**Fig 1. Effects of CGA on renal biochemical markers.**

## 3.3. CGA inhibited NLRP3 inflammasome and activated the Nrf2 pathway in high-fat diet (HFD)/ streptozotocin (STZ) -induced diabetic kidneys

In vivo results showed that expression of NLRP3 protein was remarkably increased in the DM group, and similar changes in cleaved caspase-1(c-caspase-1), IL-1β and IL-18 were consistently observed. However, these increases were significantly inhibited by CGA treatment in diabetic kidneys, as shown in Fig 3A. In immunohistochemistry, the expression of NLRP3 and ASC were significantly increased in the DM group, which further validated the activation of the NLRP3 inflammasome. Chlorogenic acid treatment can reduce the expression of NLRP3 and ASC, as show in Fig 3B. We also evaluated the effect of CGA on the Nrf2 pathway using Western blotting. The results demonstrated that nuclear Nrf2 and the Nrf2 downstream target heme oxygenase-1(HO-1) expression were decreased in the diabetic group. The expression of nuclear Nrf2 and HO-1 were enhanced in response to CGA treatment, as shown in Fig 3C. The membrane images of the immunoprotein blotting are in S1 File.

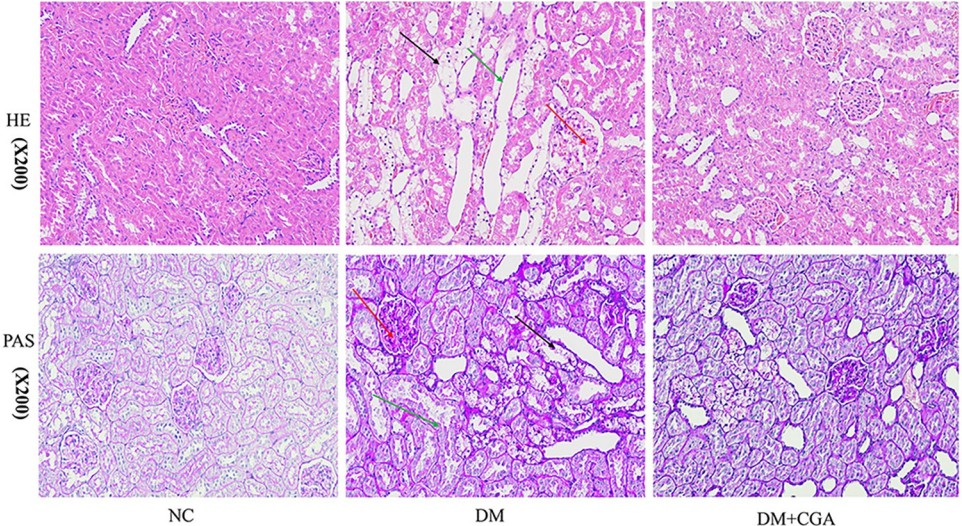

**Fig 2. The effects of CGA on renal histopathological injury.** Renal tissues were stained with haematoxylin and eosin (HE ×200) and periodic acid-Schiff (PAS ×200). The glomerular area and mesangial area were quantitatively analyzed. A total of 10 glomeruli were analyzed from each rat to calculate the mesangial area using ImageJ software.

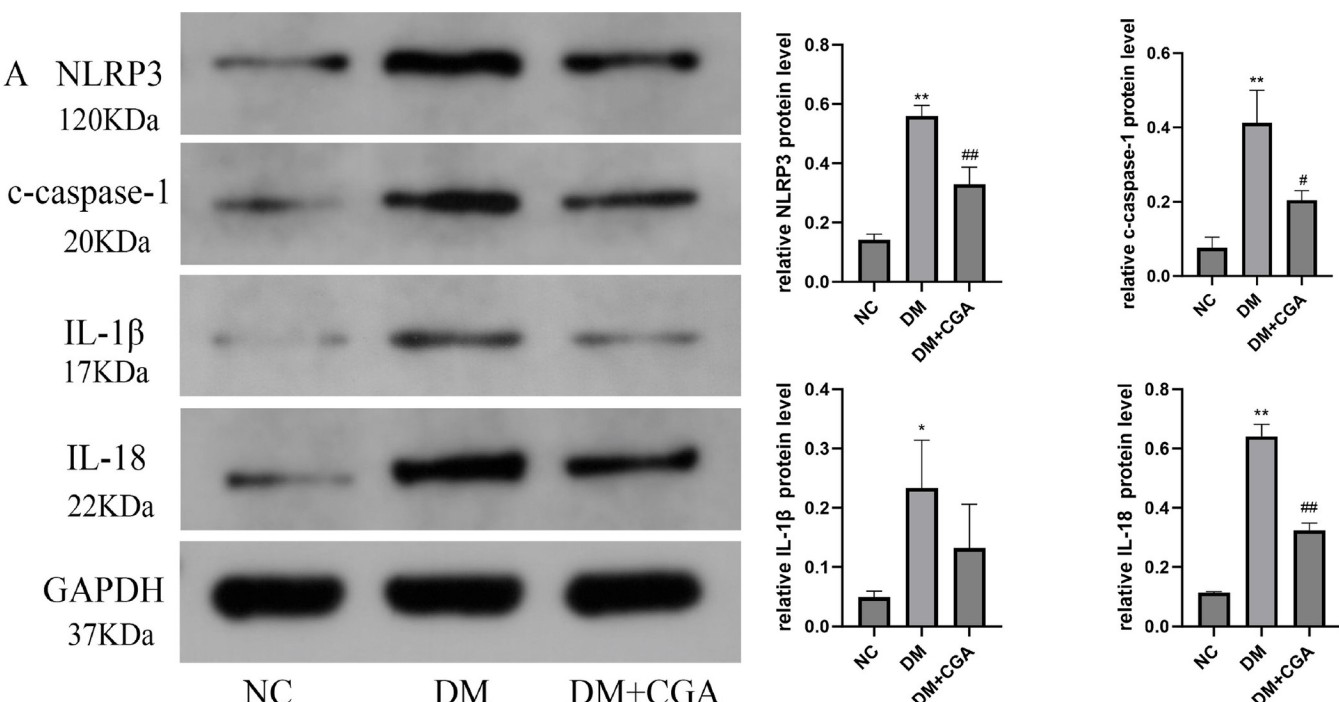

**Fig 3. The effects of CGA on NLRP3 inflammasome and the Nrf2 pathway in high-fat diet (HFD)/ streptozotocin (STZ) -induced diabetic kidneys.** A. The protein levels of NLRP3, c-caspase-1, IL-1β and IL-18 were detected by Western blotting in renal tissues. *P < 0.05, **P < 0.01 vs. NC. #P < 0.05, ##P < 0.01 vs. DM. B. The protein levels of NLRP3 and ASC were detected by immunohistochemical staining. C. The protein levels of nuclear Nrf2 and HO-1 were detected by Western blotting in renal tissues. *P < 0.05, **P < 0.01 vs. NC. #P < 0.05, ##P < 0.01 vs. DM.

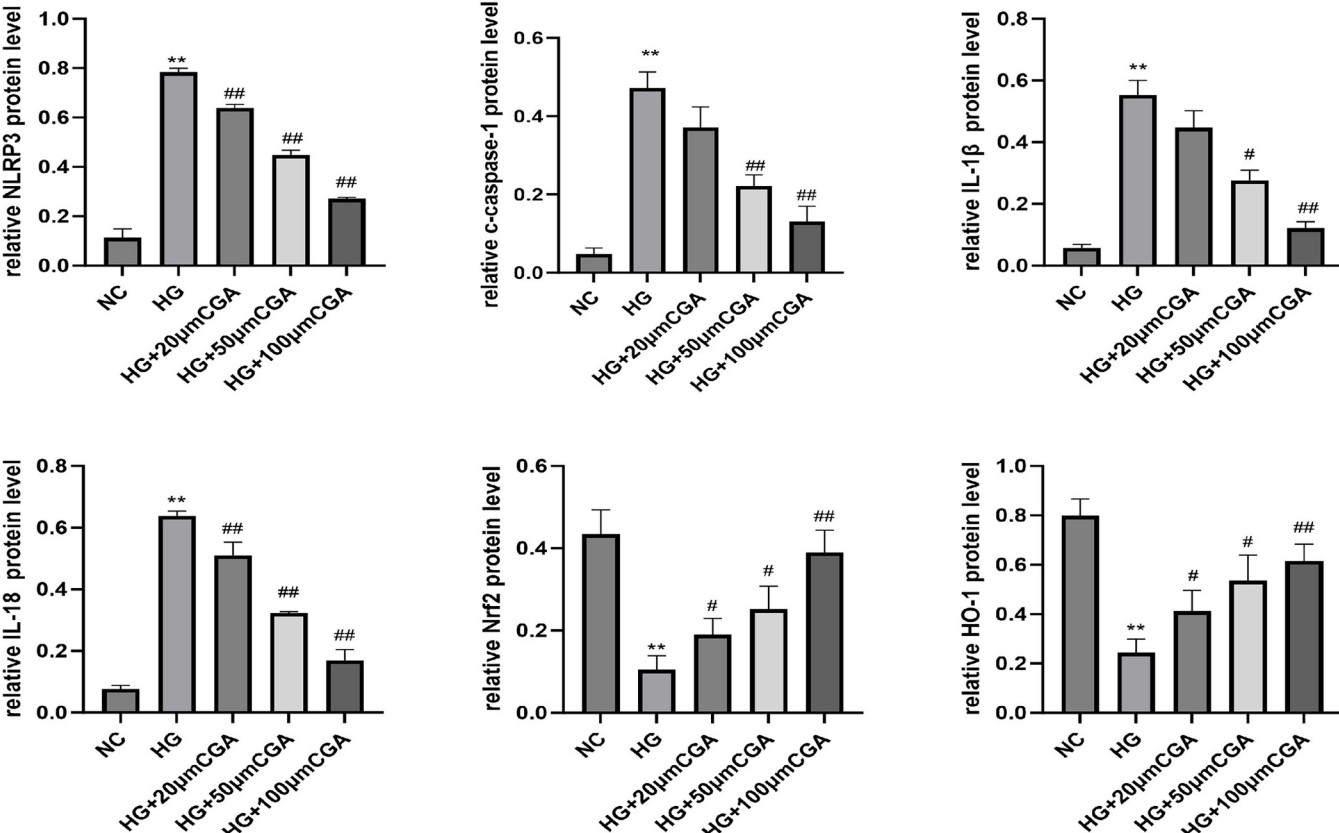

**Fig 4. The effects of CGA on NLRP3 inflammasome and the Nrf2 pathway in HK-2 cells.** A. The protein levels of NLRP3, c-caspase-1, IL-1β and IL-18 were analysed using Western blotting, and GAPDH served as a control. B. The protein levels of nuclear Nrf2 and HO-1 were detected by Western blotting. *P < 0.05, **P < 0.01 vs. NC. #P < 0.05, ##P < 0.01 vs. HG.

### 3.4. CGA inhibited NLRP3 inflammasome and activated the Nrf2 pathway in HK-2 cells

Cell experiment results showed that high glucose (HG) induced increases in the levels of NLRP3, cleaved caspase-1, IL-1β and IL-18, whereas treatment with CGA markedly decreased the levels of these NLRP3 inflammasome activation-related biomarkers, as shown in Fig 4A. Furthermore, CGA inhibited NLRP3 inflammasome activation in a concentration-dependent manner. We assessed the activation effect of CGA in vitro. As shown in Fig 4B, the expression of nuclear Nrf2 and HO-1 were decreased in the HG group. CGA increased the expression of nuclear Nrf2 and HO-1 compared with the HG group in a concentration-dependent manner.

### 3.5. CGA inhibited NLRP3 inflammasome activation through modulation of the Nrf2 pathway in HK-2 cells

To verify the effect of Nrf2 in CGA-mediated protection against HG-induced NLRP3 inflammasome activation, we knocked down Nrf2 expression using siRNA, as shown in Fig 5. The results showed that HG treatment significantly increased NLRP3 inflammasome activation in HK-2 cells, which could be down-regulated by the intervention of CGA. However, silencing of Nrf2 inhibited the effect of CGA on NLRP3 inflammasome activation induced by HG. These results demonstrated that the NLRP3 inflammasome inhibitory effect of CGA was mediated

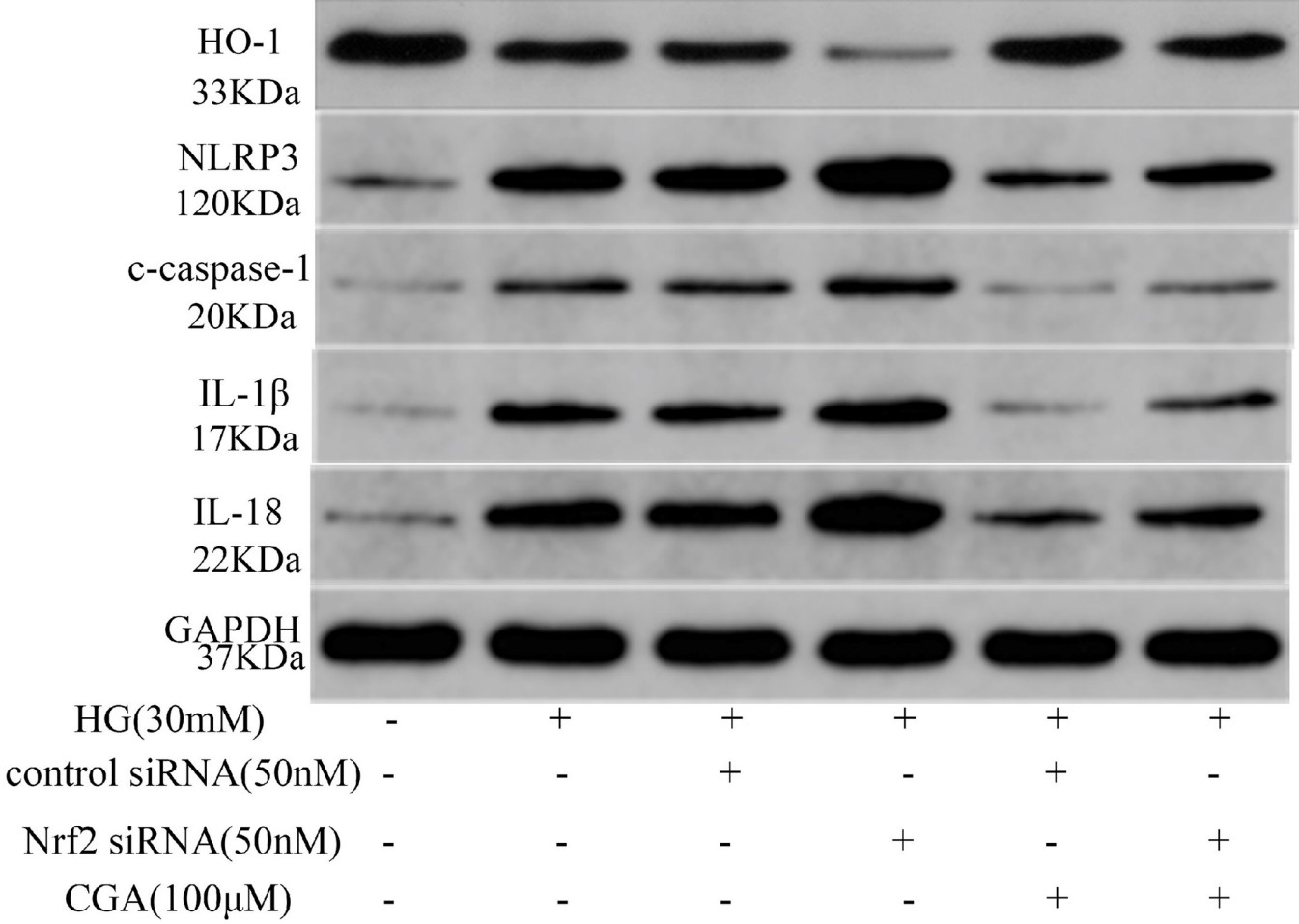

**Fig 5. Intervention in the CGA-induced activation of the Nrf2 pathway and its inhibition of NLRP3 inflammasome in HK-2 cells.** The protein expression of HO-1, NLRP3, c-caspase-1, IL-1β and IL-18 protein expression was analysed using Western blotting. *P < 0.05, **P < 0.01 vs. NC. #P < 0.05, ##P < 0.01 vs. HG. ^P < 0.05, ^^P < 0.01 vs. HG. + Control siRNA. +P < 0.05, ++P < 0.01 vs. HG. + Control siRNA + CGA.

by the Nrf2 activation at least partly. Nrf2 plays pivotal role in CGA's attenuation of DN progression through inhibiting NLRP3 inflammasome activation.

## 4. Discussion

Diabetic nephropathy (DN) is one of fatal microvascular complications of diabetes patients in morbidity and mortality. Although several available therapeutic interventions have been reported to retard the progression of DN, the high morbidity and high mortality rates of DN have not declined significantly [19]. Therefore, the novel and effective therapeutic strategies have become a hot topic around world. Growing evidence has demonstrated that some natural plant compounds or herbal products have therapeutic potential via regulation of inflammation to prevent and potentially treat DN [20–22], such as abelmoschus manihot ameliorates podocyte pyroptosis and injury in high glucose conditions by targeting NLRP3 inflammasome activation [22]. Chlorogenic acid (CGA) with a good Nrf2 activation effect possessed anti-inflammatory properties. Little information concerning CGA is available on DN, and the underlying mechanism still remains to be further revealed. In the current study, CGA obviously improved the renal function of diabetic rats. And we verified that CGA treatment

ameliorated the pathological injury in the kidney tissues of diabetic rats through the histological study.

The activation of NLRP3 inflammasome plays a pivotal role in the occurrence and progression of DN. Extensive evidence has shown that circulating inflammasome and pro-inflammatory cytokines secretion were increased in DN patients and animal models, while decreasing the activation of the inflammasome mitigates diabetic kidney damage and delays progression [23]. Hence, therapies targeting NLRP3 inflammasome activation may effectively protect renal function renal function and prevent the progression of DN. Huang X et al. [24] reported that Nrf2 and NLRP3 participated in the CGA-mediated anti-inflammatory reaction. Shi A et al. [16]suggested that CGA protected against CCl4-induced acute liver injury through inhibiting NLRP3 inflammasome activation. Zeng J et al. [25] indicated that CGA prevented colitis by inactivating the NF-κB/NLRP3 inflammasome pathway in macrophages. In this study, we observed that the NLRP3 inflammasome was activated, indicated by the increased expression of NLRP3, cleaved caspase-1(c-caspase-1), IL-1β and IL-18 in diabetic rats and HG-induced HK-2 cells. Increased expression of NLPR3 and ASC in immunohistochemistry further confirmed the activation of the NLRP3 inflammasome. While CGA obviously suppressed the NLRP3 inflammasome activation by decreasing the expression of NLRP3, c-caspase-1, IL-1β and IL-18 in diabetic rats and HG-induced HK-2 cells. Our cell experiments also showed that CGA could suppress NLRP3 inflammasome activation. These findings support that treatment with CGA may be a potential therapeutic drug for DN by inhibiting NLRP3 inflammasome activation.

In recent years, many studies have explored the possible mechanisms of NLRP3 inflammasome activation, such as flux of ions, rupture of lysosomes, generation of ROS, release of mitochondrial DNA [26]. Although the mechanisms participated in NLRP3 inflammasome activation are still not been fully elucidated, oxidative stress has been reported to play a significant role in activating NLRP3 inflammasome [27]. Nrf2 plays a crucial role in endogenous oxidative stress. Numerous studies have suggested that CGA play important roles in activating of Nrf2 and stimulating antioxidant enzymatic activities to alleviate oxidative damage. Resent studies have shown that the treatment with CGA ameliorated the expressions of Nrf2, HO-1, NLRP3, ASC and inhibited the contents of IL-1β, IL-18 [24]. Our data displayed that CGA promoted Nrf2 and its target gene HO-1 expression in vitro. CGA also promoted Nrf2 activation, and promoted the expression of downstream antioxidant enzyme HO-1in HG- induced HK-2 cells. These findings show that treatment with CGA significantly improved kidney function and reduced urinary protein excretion and the effects were associated with the Nrf2 signalling pathway activation. Zhang C et al. [28] identified that Nrf2 is the upstream regulator of NLRP3 in parkinson's disease by using three mice models of genetic defects (Nrf2-KO, NLRP3-KO and Caspase-1-KO). Another study also showed that the expression of NLRP3 was upregulated after Nrf2 silencing, while knocking down NLRP3 did not affect the expression of Nrf2 [29]. Hurtado-Navarro L et al. [30] also highlighted the potential application of Nrf2 inducers in the prevention of NLRP3 inflammasome activation. The further mechanisms are still elusive. Several studies attempted to explain the further mechanisms between the Nrf2 and the NLRP3 inflammasome [8,31]. Hou Y et al. [8]suggested that Nrf2 is a protective regulator against NLRP3 inflammasome activation by regulating the regulating thioredoxin1 (Trx1) / thioredoxin interacting protein (TXNIP) complex in cerebral ischemia reperfusion injury. Moreover, generation of ROS can promote NLRP3 inflammasome activation [26]. Nrf2 may suppress NLRP3 inflammasome activation by reducing ROS generation. In our study, silencing the Nrf2 using small interfering RNA (siRNA) increased the expression of NLRP3, c-caspase-1, IL-1β and IL-18 in HG-induced HK-2 cells. The inhibitory effect of chlorogenic acid on NLRP3 inflammasome was abolished after the intervention of Nrf2 siRNA. The results

suggesting that the Nrf2 pathway play a vital role in regulating NLRP3 inflammasome activation. And CGA inhibited NLRP3 inflammasome activation through modulation of the Nrf2 pathway in DN. However, the role of Trx1/ TXNIP complex in the regulation of NLRP3 inflammasome by Nrf2 remains to be further investigated.

In conclusion, CGA protects against diabetic kidney injury in vitro and in vivo through activation of the Nrf2 pathway and inhibition of the activation of NLRP3 inflammasome. Moreover, the CGA induced inhibitory effect on NLRP3 inflammasome activation through modulation of the Nrf2 pathway. These findings suggest that CGA is a novel therapeutic choice for the treatment of DN.

## Supporting information

**S1 Table. The biochemical data and protein expression levels.**
(XLSX)

**S1 File. The membrane images of the immunoprotein blotting.**
(PDF)

## Author Contributions

**Conceptualization:** Liping Bao, Haihui Yang.

**Data curation:** Yuhan Gong.

**Formal analysis:** Yuhan Gong.

**Funding acquisition:** Liping Bao, Haihui Yang.

**Investigation:** Liping Bao, Yuhan Gong, Wenji Xu.

**Methodology:** Liping Bao, Yuhan Gong, Wenji Xu.

**Project administration:** Haihui Yang.

**Resources:** Jun Dao, Jinjin Rao.

**Software:** Wenji Xu.

**Supervision:** Jun Dao.

**Validation:** Jun Dao.

**Visualization:** Jinjin Rao.

**Writing – original draft:** Yuhan Gong.

**Writing – review & editing:** Liping Bao, Jinjin Rao, Haihui Yang.

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
