## [Decision Letter · Decision Letter 0]

28 Aug 2024

PONE-D-24-12451Chlorogenic acid inhibits NLRP3 inflammasome activation through Nrf2 activation in diabetic nephropathyPLOS ONE

Dear Dr. Bao,

Thank you for submitting your manuscript to PLOS ONE. After careful consideration, we feel that it has merit but does not fully meet PLOS ONE’s publication criteria as it currently stands. Therefore, we invite you to submit a revised version of the manuscript that addresses the points raised during the review process.

 Specifically, please address the following points:Inclusion of complete western blot membranes with molecular weight markers important for evaluating pro- and cleaved forms of respective enzymes and cytokines. This may be included as supplementary information.Review and address concerns regarding the different dosing concentrations described throughout the manuscript.Review and address concerns regarding language some of the reviewers found confusing. For example, the statement regarding Nrf2 and its negative regulation of the NLRP3 inflammasome. You may consider addressing this by expanding the introduction and/or discussion.Address concerns regarding lack of detail in the methodology including, but not limited to, catalog numbers, dilution factors, etc.Address concerns regarding the presentation of figures including, but not limited to, inclusion of arrows to emphasize respective areas of figures, normalization of figure presentation, etc.Address recommendations regarding inclusion of NLRP3 inhibitors or NLRP3 KO mice, and potential explanations for mechanisms of NLRP3 activation of DN.

We look forward to receiving your revised manuscript.

Kind regards,

Jordan Robin Yaron, Ph.D.

Academic Editor

PLOS ONE

Journal requirements: 1. When submitting your revision, we need you to address these additional requirements. Please ensure that your manuscript meets PLOS ONE's style requirements, including those for file naming. The PLOS ONE style templates can be found at https://journals.plos.org/plosone/s/file?id=wjVg/PLOSOne_formatting_sample_main_body.pdf and https://journals.plos.org/plosone/s/file?id=ba62/PLOSOne_formatting_sample_title_authors_affiliations.pdf. 2. We note that the grant information you provided in the ‘Funding Information’ and ‘Financial Disclosure’ sections do not match.  When you resubmit, please ensure that you provide the correct grant numbers for the awards you received for your study in the ‘Funding Information’ section. 3. Thank you for stating the following in the Acknowledgments Section of your manuscript: [The present study was supported by Kunming Medical University applications of Yunnan (grant number 202101AY070001-304 to Liping Bao]We note that you have provided funding information that is not currently declared in your Funding Statement. However, funding information should not appear in the Acknowledgments section or other areas of your manuscript. We will only publish funding information present in the Funding Statement section of the online submission form. Please remove any funding-related text from the manuscript and let us know how you would like to update your Funding Statement. Currently, your Funding Statement reads as follows:  [The author(s) received no specific funding for this work.] Please include your amended statements within your cover letter; we will change the online submission form on your behalf. 4. Please provide a complete Data Availability Statement in the submission form, ensuring you include all necessary access information or a reason for why you are unable to make your data freely accessible. If your research concerns only data provided within your submission, please write "All data are in the manuscript and/or supporting information files" as your Data Availability Statement. 5. PLOS ONE now requires that authors provide the original uncropped and unadjusted images underlying all blot or gel results reported in a submission’s figures or Supporting Information files. This policy and the journal’s other requirements for blot/gel reporting and figure preparation are described in detail at https://journals.plos.org/plosone/s/figures#loc-blot-and-gel-reporting-requirements and https://journals.plos.org/plosone/s/figures#loc-preparing-figures-from-image-files. When you submit your revised manuscript, please ensure that your figures adhere fully to these guidelines and provide the original underlying images for all blot or gel data reported in your submission. See the following link for instructions on providing the original image data: https://journals.plos.org/plosone/s/figures#loc-original-images-for-blots-and-gels.   In your cover letter, please note whether your blot/gel image data are in Supporting Information or posted at a public data repository, provide the repository URL if relevant, and provide specific details as to which raw blot/gel images, if any, are not available. Email us at plosone@plos.org if you have any questions.

Reviewers' comments:

Reviewer's Responses to Questions

**Comments to the Author**

1. Is the manuscript technically sound, and do the data support the conclusions?

Reviewer #1: Partly

Reviewer #2: Yes

Reviewer #3: Yes

Reviewer #4: Partly

2. Has the statistical analysis been performed appropriately and rigorously? 

Reviewer #1: No

Reviewer #2: Yes

Reviewer #3: Yes

Reviewer #4: Yes

3. Have the authors made all data underlying the findings in their manuscript fully available?

Reviewer #1: Yes

Reviewer #2: Yes

Reviewer #3: Yes

Reviewer #4: Yes

4. Is the manuscript presented in an intelligible fashion and written in standard English?

Reviewer #1: Yes

Reviewer #2: Yes

Reviewer #3: Yes

Reviewer #4: Yes

5. Review Comments to the Author

Reviewer #1: This paper was mainly explored through two dimensions in vivo and in vitro. In vivo, the diabetic model rats were experimentally verified by ELISA, HE staining, PAS staining and Western blot. In vitro, HK-2 cells were used to explore the protein expression changes of inflammasome and Nrf2 pathway by Western blot, and the expression changes of inflammasome and Nrf2 pathway proteins were explored by Western blot after siRNA knockdown of Nrf2 expression, so as to verify that chlorogenic acid inhibits the activation of NLRP3 inflammasome in diabetic nephropathy by activating Nrf2.

The completion of this paper is good, the basic indicators have been verified, and the main problems are that the experimental grouping is thin, and the experimental design and the use of experimental methods are relatively simple and not novel enough. The content is relatively routine, and the verified mechanism has become universal. The main issues of the manuscript are as follows:

1. The weight of the Chinese drug group in Fig1 B was higher than that of the model, which was inconsistent with the description of the results.

2. It is recommended that the WB section provide the whole membrane；

3. Only one dose was used in the treatment group, and the angle was relatively simple. There was no experimental attempt to discuss the correlation between the drug dose and the treatment effect.

4. The content of the experimental design is relatively modeled, not novel enough, and there are some bright spots lacking in technology, so it is recommended to enrich the experimental design.

Reviewer #2: Notes on Review for PLOS One

“Chlorogenic acid inhibits NLRP3 inflammasome activation through Nrf2 activation in diabetic nephropathy”

Accept with Minor Revisions

Review Comments to Author:

This manuscript is technically sound, and the data supports all conclusions presented in the final document. Statistical Analysis has been performed and indicates that the results presented are statistically significant. The authors have also made it clear that all data underlying the findings in the manuscript are fully available.

Summary:

Diabetic neuropathy is a serious complication of both type 1 and type 2 diabetes, and it often affects the kidneys’ ability to remove waste products and extra fluid from the body. Inflammation caused by the NLRP3 inflammasome and further Nrf2 activation is a large contributing factor to the progression of diabetic nephropathy (DN). Chlorogenic acid (CGA) could activate Nrf2, which is found in nature. This study evaluated the ability of CGA to reduce inflammation. Findings showed that serum creatinine (Scr), blood urea nitrogen (BUN), and urinary protein excretions decreased after applying CGA. This occurs because CGA can activate the function of Nrf2 and halt the activity of NLRP3. Thus, this study demonstrated that chlorogenic acid could decrease DN progression and showed further that this impact is largely due to suppression of the NLRP3 inflammation and is modulated through the Nrf2 pathway. This research represents potential therapeutic implications for DN.

Abstract:

Comments: Overall, the abstract explains the background and premise of the article concisely and fully.

Major Revisions:

Minor Revisions:

1. In the abstract, you wrote: “CGA administration can active the Nrf2 pathway and inhibited NLRP3 inflammasome activation.” However, it should be noted that “CGA administration can activate the Nrf2 pathway and inhibit NLRP3 inflammasome activation.”

Introduction:

Comments:

The introduction concisely explains DN and how NLRP3 plays a role in developing the disease. It provides an adequate basis for the overall study.

Major Revisions:

1. The hypothesis should be rephrased; the way it is currently explained it does not explain how the Nrf2 negatively regulates NLRP3 inflammasome, particularly in this segment “negatively regulated the activation of NLRP3 inflammasome via regulation of the NLRP3 inflammasome.”

Minor Revisions:

None

Methods:

The methods described here were sufficient

Major Revisions: None

Minor Revisions:

I see that there are statistics performed on each of these experiments, but I could not find where there was a mention of how many times these experiments were completed. If they were done in duplicate or triplicate, please state this in the methods, particularly for the western blot analysis results and subsequent relative protein quantification.

Results:

Major Revisions: None

Minor Revisions: None

Though this is minor, I would make sure that all figures are uniform, there are a few that do not have the same borders which may make interpreting where certain bars are on the y axis confusing.

I think it was wise to quantify the relative levels of protein in western blot analysis.

Discussion:

Major Revisions: None

Minor Revisions:

1. The discussion mentions that “Growing evidence has demonstrated that some natural plant compounds or herbal products have therapeutic potential via regulation of inflammation to prevent and potentially treat DN[20-22]” but does not provide further details about where these compounds are found. This section would be improved in examples of where these compounds are found were also included.

2. Some small grammar repairs, consider changing “Chlorogenic acid (CGA) with a good Nrf2 activation effect possessed anti-inflammatory property. Little information concerning CGA is available on DN, and the underlying mechanism still remain to be further revealed.”

To “Chlorogenic acid (CGA) with a good Nrf2 activation effect possessed anti-inflammatory properties. Little information concerning CGA is available on DN, and the underlying mechanism still remains to be further revealed.”

Reviewer #3: The primary claim made by the authors of this submission is that the administration of Chlorogenic acid (CGA) demonstrates a renoprotective effect in the context of Diabetic Neuropathy (DN). Further the authors posit this effect is partially mediated through down-regulation of the NLRP3 inflammasome activation via the Nrf2 pathway. Reduction in the progression of kidney damage after treatment with CGA is significant due high rates of mortality despite treatment.

The authors cite multiple studies demonstrating the role inflammation reactions play in the development of DN as well as their own previous work demonstrating increased renal Nrf2 expression following CGA treatment.

Based on previous literature reports and their own previous studies on CGA treatment, the authors hypothesize that “Nrf2 negatively regulated the activation of NLRP3 inflammasome via regulation of the NLRP3 inflammasome in DN.” This statement is not written clearly and adds confusion to the mechanism the authors are targeting. The hypothesis of any research study should be clear and unambiguous as to the authors intended meaning. This statement should be re-written to explain their reasoning and scientific justification more clearly.

The authors used both an in-vitro model of cells supplemented with high levels of glucose as well as a high-fat/STZ induced diabetic rat model to replicate the Diabetes Mellitus condition. They clearly demonstrated their intended targets of study associated with NLRP3 inflammasome mechanism of action such as caspase-1, IL-1b, and IL-18. Additionally, they have identified a downstream protein of interest, heme oxgenase-1 (HO-1), which is associated with the Nrf2 pathway activation. The significance of these targets to clinical disease and oxidative stress were supported by previous literature. The methods associated with the experimental assays – histology, western blot, and blood and urine chemistries - were detailed and informative to allow for experiments to be reproduced.

Most of the results and data figures were clearly labeled and supported the conclusions stated by the authors. However, in section 3.1 the authors state that diabetic rats treated with CGA [DM + CGA group] had a lower body weight compared to untreated diabetic rats [DM group]. This is result is not what is represented in figure 1B – the DM + CGA treatment group are shown to have a higher body weight compared to the DM group. This result should be re-written to match the corresponding figure.

Overall, I felt this was a strong manuscript with a clearly defined objective and orthogonally designed experiments. The conclusions drawn by the authors are supported by the results and delve into the possible mechanisms associated with the benefits of CGA treatment in Diabetes Mellitus and inflammatory reactions. The hypothesis should be revised to clearly state their goal as well as address the conclusion drawn from figure 1B.

Reviewer #4: In this manuscript, Bao et al. explore the effects of chlorogenic acid (CGA) on diabetic nephropathy (DN). Using a diabetic rat model, they find that CGA has potential therapeutic implications for slowing the progression of DN by activating the Nrf2 pathway and inhibiting NLRP3 inflammasome activation. However, the data presented in the manuscript are sometimes not convincing and lack proper description. Addressing the following comments could help improve the quality of this study:

1. It would be better if providing histological quantification and using arrowheads to indicate pathological damage in Figure 2.

2. The catalog numbers of the antibodies used in the manuscript are not provided.

In all Western blot images, it is unclear whether IL-1β, IL-18, and caspase-1 proteins are in their pro or cleaved forms, since cleaved caspase-1 and cytokines (IL-1β, IL-18) are indicators of NLRP3 inflammasome activation. Providing additional data, such as ASC oligomerization or immunohistochemistry for NLRP3 and ASC proteins in the kidney or cells, could offer more solid evidence for NLRP3 activation. Alternatively, it would be best to use NLRP3-deficient mice to determine if NLRP3 is truly involved in the progression of DN would strengthen the findings. If knockout mice are unavailable, NLRP3 inhibitors could be used.

3. Please include molecular weight markers in all Western blot images.

4. The NLRP3 inflammasome activates the downstream effector caspase-1, leading to caspase-1-mediated pyroptosis. Since the authors have identified NLRP3 activation under their experimental conditions, it would be interesting to investigate whether CGA could prevent DN by inhibiting NLRP3 inflammasome activation and subsequently reducing pyroptosis,

5. The discussion section could be improved by exploring possible mechanisms of NLRP3 activation in DN and how Nrf2 is induced as part of the defense system against oxidative stress, as well as how NLRP3 inhibition occurs under these conditions.

6. PLOS authors have the option to publish the peer review history of their article (what does this mean?). If published, this will include your full peer review and any attached files.

Reviewer #1: No

Reviewer #2: No

Reviewer #3: **Yes: **Nicole Grigaitis-Esman

Reviewer #4: No

---

## [Author Response · Author response to Decision Letter 0]

9 Nov 2024

Reviewer #1: 

1. The weight of the drug group in Fig1 B was higher than that of the model, which was inconsistent with the description of the results.

Response: Thank you very much for your careful review and pointing out this issue. Kidney weight of diabetic rats was significantly higher than normal rats, diabetic rats treated with CGA had lower kidney weight than untreated diabetic rats. However, body weight of diabetic rats was significantly lower than normal rats, diabetic rats treated with CGA had higher body weight than untreated diabetic rats. We have revised this description in the resubmitted manuscript.

2. It is recommended that the WB section provide the whole membrane；

Response: Thank you very much for your suggestion, we have submitted the membranes for three biological replicates of WB.

3. Only one dose was used in the treatment group, and the angle was relatively simple. There was no experimental attempt to discuss the correlation between the drug dose and the treatment effect.

Response: Thank you very much for your suggestion. Because of the difficulties in establishing the diabetic rat model, the long duration of drug intervention, and the high cost, we only used one dose. To remedy this limitation, we set up different drug doses(20uM,50uM,100uM) in the cell experiment to discuss the relationship between drug dose and treatment effect. The result showed CGA inhibi NLRP3 inflammasome activation and increased the expression of nuclear Nrf2 and HO-1in a concentration-dependent manner. These mean that the treatment effect of CGA is closely related to its dose.

4. The content of the experimental design is relatively modeled, not novel enough, and there are some bright spots lacking in technology, so it is recommended to enrich the experimental design.

Response: Thank you very much for your suggestion. NLRP3 inflammasome is composed of NLRP3, caspase-1 and apoptosis-associated speck-like protein containing a C-terminal caspase recruitment domain (ASC). In the diabetic state, NLRP3 inflammasome is activated, then NLRP3 recruits ASC and cleaves caspase-1. We demonstrated the effect of chlorogenic acid on NLRP3 inflammasome by immunohistochemistry for NLRP3 and ASC proteins.

Reviewer #2: 

Abstract:

In the abstract, you wrote: “CGA administration can active the Nrf2 pathway and inhibited NLRP3 inflammasome activation.” However, it should be noted that “CGA administration can activate the Nrf2 pathway and inhibit NLRP3 inflammasome activation.”

Response: Thank you very much for your careful review and pointing out this issue. We have revised this description in the resubmitted manuscript.

Introduction:

The hypothesis should be rephrased; the way it is currently explained it does not explain how the Nrf2 negatively regulates NLRP3 inflammasome, particularly in this segment “negatively regulated the activation of NLRP3 inflammasome via regulation of the NLRP3 inflammasome.”

Response: Thank you very much for your careful review and pointing out this issue. We have revised this description in the resubmitted manuscript.

Methods:

I see that there are statistics performed on each of these experiments, but I could not find where there was a mention of how many times these experiments were completed. If they were done in duplicate or triplicate, please state this in the methods, particularly for the western blot analysis results and subsequent relative protein quantification.

Response: Thank you very much for your suggestion, all experimental data are based on three biological replicates, we have further clarified this issue in Statistical analysis.

Results:

Though this is minor, I would make sure that all figures are uniform, there are a few that do not have the same borders which may make interpreting where certain bars are on the y axis confusing.

I think it was wise to quantify the relative levels of protein in western blot analysis.

Response: Thank you very much for your suggestion. We have modified the figures in the resubmitted manuscript.

Discussion:

1. The discussion mentions that “Growing evidence has demonstrated that some natural plant compounds or herbal products have therapeutic potential via regulation of inflammation to prevent and potentially treat DN[20-22]” but does not provide further details about where these compounds are found. This section would be improved in examples of where these compounds are found were also included.

Response: Thank you very much for your suggestion. In the discussion section, we have added the example of abelmoschus manihot. Abelmoschus manihot ameliorates podocyte pyroptosis and injury in high glucose conditions by targeting NLRP3 inflammasome activation.

2. Some small grammar repairs, consider changing “Chlorogenic acid (CGA) with a good Nrf2 activation effect possessed anti-inflammatory property. Little information concerning CGA is available on DN, and the underlying mechanism still remain to be further revealed.” to “Chlorogenic acid (CGA) with a good Nrf2 activation effect possessed anti-inflammatory properties. Little information concerning CGA is available on DN, and the underlying mechanism still remains to be further revealed.”

Response: Thank you very much for your careful review and pointing out this issue. We have revised this description in the resubmitted manuscript.

Reviewer #3: The primary claim made by the authors of this submission is that the administration of Chlorogenic acid (CGA) demonstrates a renoprotective effect in the context of Diabetic Neuropathy (DN). Further the authors posit this effect is partially mediated through down-regulation of the NLRP3 inflammasome activation via the Nrf2 pathway. Reduction in the progression of kidney damage after treatment with CGA is significant due high rates of mortality despite treatment.

The authors cite multiple studies demonstrating the role inflammation reactions play in the development of DN as well as their own previous work demonstrating increased renal Nrf2 expression following CGA treatment.

Based on previous literature reports and their own previous studies on CGA treatment, the authors hypothesize that “Nrf2 negatively regulated the activation of NLRP3 inflammasome via regulation of the NLRP3 inflammasome in DN.” This statement is not written clearly and adds confusion to the mechanism the authors are targeting. The hypothesis of any research study should be clear and unambiguous as to the authors intended meaning. This statement should be re-written to explain their reasoning and scientific justification more clearly.

The authors used both an in-vitro model of cells supplemented with high levels of glucose as well as a high-fat/STZ induced diabetic rat model to replicate the Diabetes Mellitus condition. They clearly demonstrated their intended targets of study associated with NLRP3 inflammasome mechanism of action such as caspase-1, IL-1b, and IL-18. Additionally, they have identified a downstream protein of interest, heme oxgenase-1 (HO-1), which is associated with the Nrf2 pathway activation. The significance of these targets to clinical disease and oxidative stress were supported by previous literature. The methods associated with the experimental assays – histology, western blot, and blood and urine chemistries - were detailed and informative to allow for experiments to be reproduced.

Most of the results and data figures were clearly labeled and supported the conclusions stated by the authors. However, in section 3.1 the authors state that diabetic rats treated with CGA [DM + CGA group] had a lower body weight compared to untreated diabetic rats [DM group]. This is result is not what is represented in figure 1B – the DM + CGA treatment group are shown to have a higher body weight compared to the DM group. This result should be re-written to match the corresponding figure.

Overall, I felt this was a strong manuscript with a clearly defined objective and orthogonally designed experiments. The conclusions drawn by the authors are supported by the results and delve into the possible mechanisms associated with the benefits of CGA treatment in Diabetes Mellitus and inflammatory reactions. The hypothesis should be revised to clearly state their goal as well as address the conclusion drawn from figure 1B.

Response: Thank you very much for your careful review and positive comments on this study. “Nrf2 negatively regulated the activation of NLRP3 inflammasome via regulation of the NLRP3 inflammasome in DN.”- this statement is not written clearly. We have modified it to “Nrf2 negatively regulated the activation of NLRP3 inflammasome in DN.”. Our explanation in section 3.1 contains errors. Kidney weight of diabetic rats was significantly higher than normal rats, diabetic rats treated with CGA had lower kidney weight than untreated diabetic rats. However, body weight of diabetic rats was significantly lower than normal rats, diabetic rats treated with CGA had higher body weight than untreated diabetic rats. We have made revisions in the resubmitted manuscript. Thank you very much for pointing out the errors in our manuscript.

Reviewer #4: In this manuscript, Bao et al. explore the effects of chlorogenic acid (CGA) on diabetic nephropathy (DN). Using a diabetic rat model, they find that CGA has potential therapeutic implications for slowing the progression of DN by activating the Nrf2 pathway and inhibiting NLRP3 inflammasome activation. However, the data presented in the manuscript are sometimes not convincing and lack proper description. Addressing the following comments could help improve the quality of this study:

1. It would be better if providing histological quantification and using arrowheads to indicate pathological damage in Figure 2.

Response: Thank you very much for your careful review and suggestion. We have marked the pathological damages with arrows and quantified the glomerular area and mesangial area.

2. The catalog numbers of the antibodies used in the manuscript are not provided.

Response: Thank you very much for your careful review and suggestion. We have provided the catalog numbers of the antibodies in 2.1. Reagents.

In all Western blot images, it is unclear whether IL-1β, IL-18, and caspase-1 proteins are in their pro or cleaved forms, since cleaved caspase-1 and cytokines (IL-1β, IL-18) are indicators of NLRP3 inflammasome activation. Providing additional data, such as ASC oligomerization or immunohistochemistry for NLRP3 and ASC proteins in the kidney or cells, could offer more solid evidence for NLRP3 activation. Alternatively, it would be best to use NLRP3-deficient mice to determine if NLRP3 is truly involved in the progression of DN would strengthen the findings. If knockout mice are unavailable, NLRP3 inhibitors could be used.

Response: Thank you very much for your careful review and pointing out this issue. In the introduction, we mentioned “cleaved caspase-1 promotes the conversion of inflammatory factors such as pro-IL-1β and pro-IL-18 into IL-1β and IL-18 and released outside the cell, which leading to pro-inflammatory responses”. So, IL-1β and IL-18 are their cleaved forms in Western blot images. Capase-1 is its cleaved form in WB images. And we have modified capase-1 to c-capase-1 (cleaved capase-1) in all Western blot images. According to your suggestion, we performed immunohistochemistry for NLRP3 and ASC proteins in the kidney to offer more solid evidence for NLRP3 activation.

3. Please include molecular weight markers in all Western blot images.

Response: Thank you very much for your careful review and suggestion. We have included molecular weight markers in all Western blot images in revised manuscript.

4. The NLRP3 inflammasome activates the downstream effector caspase-1, leading to caspase-1-mediated pyroptosis. Since the authors have identified NLRP3 activation under their experimental conditions, it would be interesting to investigate whether CGA could prevent DN by inhibiting NLRP3 inflammasome activation and subsequently reducing pyroptosis.

Response: Thank you very much for your careful review and suggestion. As you said, it's interesting to investigate whether CGA could prevent DN by inhibiting NLRP3 inflammasome activation and subsequently reducing pyroptosis. The manuscript only describes caspase-1 dependent pyroptosis. In our next step of research, we will explore the effect of chlorogenic acid on pyroptosis and other possible mechanisms.

5. The discussion section could be improved by exploring possible mechanisms of NLRP3 activation in DN and how Nrf2 is induced as part of the defense system against oxidative stress, as well as how NLRP3 inhibition occurs under these conditions.

Response: Thank you for your suggestion. In recent years, many studies have explored the possible mechanisms of NLRP3 inflammasome activation, such as flux of ions, rupture of lysosomes, generation of ROS, release of mitochondrial DNA. Our previous study has shown that pre-treatment with CGA increased the renal expression of Nrf2 and the downstream target heme oxygenase-1 (HO-1). Hou Y et al. suggested that Nrf2 is a protective regulator against NLRP3 inflammasome activation by regulating the regulating thioredoxin1 (Trx1) / thioredoxin interacting protein (TXNIP) complex in cerebral ischemia reperfusion injury. Moreover, generation of ROS can promote NLRP3 inflammasome activation. Nrf2 may suppress NLRP3 inflammasome activation by reducing ROS generation. We also conducted relevant discussion in the discussion section.

---

## [Decision Letter · Decision Letter 1]

15 Dec 2024

Chlorogenic acid inhibits NLRP3 inflammasome activation through Nrf2 activation in diabetic nephropathy

PONE-D-24-12451R1

Dear Dr. Yang,

We’re pleased to inform you that your manuscript has been judged scientifically suitable for publication and will be formally accepted for publication once it meets all outstanding technical requirements.

Kind regards,

Jordan Robin Yaron, Ph.D.

Academic Editor

PLOS ONE

Additional Editor Comments (optional):

Reviewers' comments:

Reviewer's Responses to Questions

**Comments to the Author**

1. If the authors have adequately addressed your comments raised in a previous round of review and you feel that this manuscript is now acceptable for publication, you may indicate that here to bypass the “Comments to the Author” section, enter your conflict of interest statement in the “Confidential to Editor” section, and submit your "Accept" recommendation.

Reviewer #3: All comments have been addressed

Reviewer #4: All comments have been addressed

2. Is the manuscript technically sound, and do the data support the conclusions?

Reviewer #3: Yes

Reviewer #4: Yes

3. Has the statistical analysis been performed appropriately and rigorously? 

Reviewer #3: Yes

Reviewer #4: Yes

4. Have the authors made all data underlying the findings in their manuscript fully available?

Reviewer #3: Yes

Reviewer #4: Yes

5. Is the manuscript presented in an intelligible fashion and written in standard English?

Reviewer #3: Yes

Reviewer #4: Yes

6. Review Comments to the Author

Reviewer #3: The authors have addressed all of my comments and recommendations in initial review, including rewording their hypothesis to convey a more concise statement and addressing the errors in Figure 3.1.

Reviewer #4: The molecular weight of cleaved caspase-1 is not 45 kDa, as stated. The authors may need to review and correct this information to ensure accuracy.

7. PLOS authors have the option to publish the peer review history of their article (what does this mean?). If published, this will include your full peer review and any attached files.

Reviewer #3: **Yes: **Nicole A. Grigaitis

Reviewer #4: No

---

## [Editor Report · Acceptance letter]

26 Dec 2024

PONE-D-24-12451R1 

PLOS ONE

Dear Dr. Yang, 

I'm pleased to inform you that your manuscript has been deemed suitable for publication in PLOS ONE. Congratulations! Your manuscript is now being handed over to our production team.

Kind regards, 

on behalf of

Dr. Jordan Robin Yaron 

Academic Editor

PLOS ONE